# The Inheritance Pattern of Key Desirable Agronomic and Fruit Quality Traits in Elite Red Papaya Genotypes

Fawad Ali [1],*, Chutchamas Kanchana-udomkan [1,2] and Rebecca Ford [1],*

1. Centre for Planetary Health and Food Security, Griffith University, Nathan Campus, Brisbane, QLD 4111, Australia
2. Department of Horticulture, Faculty of Agriculture at Kamphaeng Saen, Kasetsart University, Nakhon Pathom 73140, Thailand
* Correspondence: fawad.ali@griffith.edu.au (F.A.); rebecca.ford@griffith.edu.au (R.F.)

**Abstract:** Knowledge of the heritability, genetic advance, and stability of key traits, such as the height to the first fruit, trunk circumference, number of marketable fruit, wasted fruit, fruit weight, fruit width, fruit length, flesh thickness, cavity width, cavity length, and soluble solid contents, is required. These were determined in ten advanced generation red papaya recombinant inbred lines (RIL; F5 generation). The F5 RIL were grown in four field sites, two each within two distinct agroecological climates: the Tablelands and Coastal regions. At each site, biological replicates (trees) of each RIL and the industry-standard red papaya cultivar, RB1, were grown. Agronomic traits and fruit-specific traits were assessed at five and 10 months, respectively, after seedling transplantation to the field. Height to first fruit, trunk circumference, fruit weight, and soluble solid contents were highly heritable and stable at all field sites ($h^2_{b.s.}$, 0.7–0.9) with genetic gains of almost 18% observed for height to first fruit and fruit weight. Across all sites, the trunks of the F5 lines were 37% wider, the trees set fruit 47% closer to the ground and had 20% more marketable fruit with 33% smaller fruit cavity widths, and their fruit was 11% heavier and 12% sweeter than RB1.

**Keywords:** papaya; selective breeding; agronomic features; fruit quality; heritability; REML; BLUPs; genetic advance; trait gain percentage

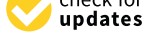



## 1. Introduction

Australian papaya (*Carica papaya* L.) is mainly grown in the Tablelands and Coastal regions of Tropical North Queensland, where 85% of the papaya industry is located [1] (p. 173). The total production volume of Australian papaya in 2021 was 19,481 tonnes with a net worth of $40.1 m AUD [1] (p. 171). In the same year, just five tonnes of Australian papaya were exported, to locations including Singapore, Hong Kong, New Zealand, United Arab Emirates, and Kuwait [1] (p. 175). Overall, the industry in Australia has experienced a 28% net value increase over the past five years, in contrast to the Hawaiian papaya industry, which had a 65% decrease in production over the 2016–2020 period [2] (p. 1). Despite the steady growth in Australian papaya production, there remains significant scope to develop new varieties that better match industry requirements and national and international consumer preferences.

Significant progress towards elite varieties may be made through selective breeding approaches and focusing on understanding the genetic mechanisms underpinning key desirable traits within bi-parental mapping populations [3]. For this, trait stabilization within the breeding material is crucial to ensure that the material that is eventually released is consistent in terms of agronomic, productivity and fruit quality. To determine genetic stabilization and gains made within the breeding program, an understanding of the heritability of key traits within diverse environments is required [4–7]. Environmental impacts on phenotypic trait expression may be considerable and undermine trait stability,

particularly if selections are not performed in the range of agroecological environmental conditions used for production [8,9]

　　　Traits with higher heritability are underpinned largely by genetic components and their expression may be less impacted by environmental factors [10]. The heritability of agronomic and fruit quality traits was calculated widely for fruit crops, including apple (*Malus pomila*) [11], citrus (*Citrus* spp.) [12], and avocado (*Persea americana*) [13]. High heritability (>80%) was previously reported in papaya for tree height [14], soluble solid contents [15], and fruit weight, length, and width [16]. However, no prior reports exist on potential environmental impacts on trait stability or heritability of other important papaya traits.

　　　In the current study, the heritability and potential environmental impacts on the stability of expression of key agronomic and fruit quality traits of papaya were investigated in two agro-geographical and climatically distinct Australian growing regions, namely the Tablelands and Coastal regions. The traits assessed included 'height to first fruit' on the fruit column from ground level, since trees that produce the initial fruit lower to the ground require less mechanized intervention for picking over the productive lifespan of the tree [17]. Moreover, since papaya trees are shallow-rooted and generally have narrow trunks, they are susceptible to windbreak [18], and therefore the 'trunk circumference' trait was also assessed. Additionally, the important traits of number of marketable and wasted fruit per tree fruit column were assessed, with the goal of understanding the potential to improve marketable fruit yield. Related to yield and consumer fruit quality preferences, the heritability and stability of 'fruit weight', 'fruit size' (width and length), 'flesh thickness', 'cavity size' (width and length), and 'sweetness' (soluble solid contents, °Brix) were also assessed. Subsequently, the best performing inbred lines in each growing region were selected.

## 2. Methods

### 2.1. Germplasm and Trial Sites

　　　Seeds of ten F5 recombinant inbred lines (RIL) of red papaya (*Carica papaya* L) that were developed within the Australian National Papaya Breeding Program led by Griffith University (Hort Innovation PP18000) were used in this study. The RIL, T3-5-6.10, T1-5-2.3, T1-5-5.9, T2-5-5.27, T2-5-3.12, C1-5-4.1, C1-5-4.2, C1-5-4.3, C3-3-5.24, and C2-5-5, were derived from a 'Solo' and 'Holland' cross in Mareeba, Queensland (16.9796° S, 145.3314° E). Single seed descended lines were selected and subsequently grown and assessed at four trial sites in Tropical North Queensland, Australia; two in the Tablelands (T1; 16.9796° S. 145.3314° E and T2; 17.16837° S. 145.11285° E) and two in the Coastal (C1; 17.063801° S. 145.96268° E and C2; 17.66115° S. 146.04259° E) region. The Tablelands and Coastal regions differed significantly in environmental and climatic factors, including in soil type of sandy loam (Tablelands) or clay (Coastal), mean annual rainfall of 826 mm (tablelands) or 3549 mm (Coastal), and average temperature ranges of 22–29 °C (Tablelands) or 19–28 °C (Coastal) [19]. The F5 lines were selected based on superior agronomic and fruit quality traits and planted in replicated rows alongside the industry red papaya standard variety, RB1, in January 2021.

### 2.2. Seedling Cultivation and Sex Determination

　　　Prior to sowing, F5 RIL seeds were soaked in 2 mM gibberellic acid solution for 30 min at 45 °C, and then rinsed with tap water twice. Soaked seeds were then individually sown into 48-cell seedling trays filled with seed-raising mix (Searles, Winya (4515), QLD-Australia). Subsequently, 100 mL of liquid fertilizer (Vigor-Lig Plus, Agmin, Australia) and 50 mL of liquid lime (PH-Plus Liquid Lime, Ultimate Products, Australia) solution was added to 9 L of water and 60 mL of the solution was applied once a week to each growing tube. Slow-release fertilizer N: P: K (15:15:15) (Lawn solutions, Australia) was added to potting mix prior to filling 48-cell seedling trays. Seedlings were grown inside a glass house with an approximate day and night temperature of 28/21 °C for 10 weeks during August

to October 2020. Additionally, the seedlings were sex genotyped from gDNA extracted from the young shoot tip at 21 days after germination [20]. The hermaphrodite seedlings were selected and then sun-hardened for three weeks prior to field transplantation.

Data were collected from six biological replicates of each RIL and RB1 at each of the trial sites at five (productivity traits) and 10 months (fruit quality traits) after seedling transplantation to the field [21]. The four productivity traits of height to the first fruit (cm), trunk circumference (cm), number of marketable fruit (counted per fruit column; refer to fully developed mature fruit), and number of wasted fruit (counted per fruit column; refers to partially developed fruit also known as carpelloid fruit) were evaluated from each tree (biological replicate). Subsequently, the entire fruit column of each tree (replicate) was visually divided into four quadrants (representing each side of the tree from the centre of the crown) and two fruit were selected randomly from each quadrant to provide a total of eight fruit per tree for assessment. Seven fruit quality traits, i.e., fruit weight (g), fruit width (cm), fruit length (cm), flesh thickness (cm), cavity width (cm), cavity length (cm), and soluble solid contents (°Brix), were assessed. The evaluation protocols for each trait were as detailed in The Papaya Evaluation Handbook (2nd edition) [21].

*2.3. Statistical Analyses*

2.3.1. Linear Mixed Model

To allow both random and fixed effects in the model and reduce noise in the collected data, linear mixed models (LMM) were fitted using restricted maximum likelihoods (REML) in DeltaGen R-Shiny-based program [22]. The least significant differences were calculated at $p < 0.05$. A best linear unbiased prediction (BLUP) approach was used to estimate the breeding value or genetic worth of each individual trait based on progeny performance. The BLUP for each trait was determined across the four trial sites, two within each of the Tablelands (T1 and T2) or Coastal (C1 and C2) regions. In the modelling, these were considered as individual "*locations*", biological replications (trees) were considered as the "*reps*", F5 RIL were considered as the "*lines*", and the commercial cultivar "RB1" was considered as the "*check*".

2.3.2. Histogram Analysis

Trait distribution histograms were constructed in R (version 3.5.2; R core team 2020) using the determined models with the "lme4" package.

2.3.3. Variance and Covariance Components and Broad-Sense Heritability

Broad-sense heritability ($H^2_{b.s}$) was calculated from the variance components of genotypic variance ($\sigma^2 g$), phenotypic variance ($\sigma^2 p$), and environmental variance ($\sigma^2 e$) for each measured trait using the *lemr*() function in the *"lme4"* package and the following equation [23]:

$$H^2_{b.s} = \sigma^2_g / \sigma^2_{ph} + \sigma^2_e$$

$$\sigma^2_g = (MSS - MSE)/r$$

$$\sigma^2_{ph} = \sigma^2_e + \sigma^2_g$$

where MSS = the mean sum of squares due to the treatment, MSE = mean sum of squares due to the environment (location), and r is the number of biological replications. The genotypic co-variance (GCV), phenotypic co-variance (PCV), and environmental co-variance (ECV) are then calculated using the following equations, where $X^-$ is the grand mean of the population:

$$PCV = 100 \times \sqrt{\sigma^2_{ph}} / X^-$$

$$GCV = 100 \times \sqrt{\sigma^2_g} / X^-$$

$$ECV = 100 \times \sqrt{\sigma^2_e} / X^-$$

### 2.3.4. Genetic Advance (Percentages)

The genetic advance (or gain) for each trait was measured as a percentage using the following equation, where k was the selection differential (k = 1.77 at 10% selection) [23,24]:

$$GA \ (\%) = 100 \times k \times h^2{}_{bs} \times \sigma^2{}_{ph}$$

### 2.3.5. Trait Gain Percentages

The potential advantage (+) or disadvantage (−) of each trait measured within each of the F5 RIL in each environment or either Tableland (T1 + T2) or Coastal-Innisfail (C1 + C2) was compared to the commercial variety RB1 and calculated as a percentage (%) using the following equation:

$$TG \ (\%) = [(Mean_{RIL} - Mean_{CH})/Mean_{CH}] \times 100 \tag{1}$$

where TG (Trait gain, %) is the increase or decrease percentage of each inbred line over the commercial variety RB1, $Mean_{RIL}$ is the mean performance of the advanced generation recombinant inbred line, and $Mean_{CH}$ is the mean performance of the commercial check variety RB1.

## 3. Results

### 3.1. Mixed Models for Exploring G × E Interaction of Measured Traits in F5 RIL

Variations in trait expressions among the F5 RIL were assessed within and among the two field sites within each region. Accordingly, no significant genotype (G) × environment (E) trait effects were determined among the F5 RIL when grown in either T1 or T2, or when grown in either C1 or C2. However, when trait variations were assessed among the same F5 RIL across the two distinct growing regions, the trait variation among the RIL grown in the Tablelands region (T1 + T2) was higher than among the RIL grown in the Coastal region (C1 + C2), which were 0–3814 and 0–2125, respectively (Table 1). The G × E interactions among regions were significant for most traits except fruit width, fruit length, and cavity length. Significant differences among regions were detected in the variation of height to first fruit (cm), trunk circumference (cm), fruit weight (g), and the number of marketable fruit ($p < 0.05$; Table 1).

**Table 1.** Linear mixed models (LMM) of key agronomic and consumer-driven fruit quality traits fitted by residual maximum likelihood (REML; $p < 0.05$). The variance components of the interactions between genotypes (G) x environments (E) were calculated in two distinct agro-geographical climates [(Tableland: T1 and T2)] or [(Coastal: C1 and C2)] in Tropical North Queensland, Australia.

| Random Factor | Traits | Among Two Field Sites within the Tablelands (T1 + T2) Environment | | Among Two Field Sites within the Coastal (C1 + C2) Environment | | Across Two Distinct Agro-Geographical Climates (Across Four Trial Sites): Tablelands (T1 and T2) and Coastal (C1 and C2) | |
|---|---|---|---|---|---|---|---|
| | | Variance | *p*-Value | Variance | *p*-Value | Variance | *p*-Value |
| | Height to the first fruit (cm) | 35.49 | 0.21 | 45.91 | 0.14 | 89.10 | 0.000 |
| | Trunk circumference (cm) | 7.14 | 0.17 | 3.21 | 0.12 | 20.01 | 0.000 |
| | Number of marketable fruit (counted) | 7.15 | 0.13 | 2.12 | 0.24 | 10.05 | 0.01 |
| Genotype among two field sites within the Coastal & Tablelands environment × Environment | Number of wasted fruit (counted) | 0.41 | 0.22 | 0.39 | 0.25 | 0.63 | 0.001 |
| | Fruit weight (g) | 3814 | 0.12 | 2125 | 0.22 | 18,424 | 0.001 |
| | Fruit width (cm) | 0.17 | 0.23 | 0.11 | 0.19 | 0.36 | 0.24 |
| | Fruit length (cm) | 0 | 0.24 | 0 | 0.17 | 0 | 0.13 |
| | Flesh thickness (cm) | 0.04 | 0.11 | 0.03 | 0.26 | 0.42 | 0.005 |
| | Cavity width (cm) | 0.08 | 0.09 | 0.02 | 0.21 | 0.47 | 0.03 |
| | Cavity length (cm) | 0.05 | 0.13 | 0.12 | 0.20 | 0.21 | 0.14 |
| | Soluble solid contents (°Brix) | 0.32 | 0.25 | 0.24 | 0.17 | 0.59 | 0.001 |

### 3.2. Heritability and Genetic Advance of Key Agronomic and Fruit Quality Traits

Several segregation patterns of inheritance were observed among the traits assessed, suggesting a unimodal monogenic inheritance of cavity width (cm), a bimodal digenic inheritance of height to first fruit (cm), number of marketable fruit, and the number of

wasted fruit, and a symmetric polygenic inheritance of trunk circumference (cm), fruit weight (g), fruit width (cm), flesh thickness (cm), cavity length (cm), fruit length (cm), and soluble solid contents (°Brix) (Figure 1).

The traits of height to first fruit (cm), trunk circumference (cm), fruit weight (g) and soluble solid contents (°Brix) were highly heritable ($h^2_{b.s.}$, 0.7–0.9; Table 2) when grown at all locations within either Tablelands or Coastal regions. Meanwhile, fruit length (cm) and cavity length (cm) were less stable, with $h^2_{b.s}$ as low as 0.04 at one location and 0.9 at another. Compared to the industry standard, RB1, substantial genetic advances of 13–18% in increased fruit weight (g) and 5–17% (g) in lowering the height to the first fruit (cm) were achieved, with significant variation among environments and trial sites (Table 3).

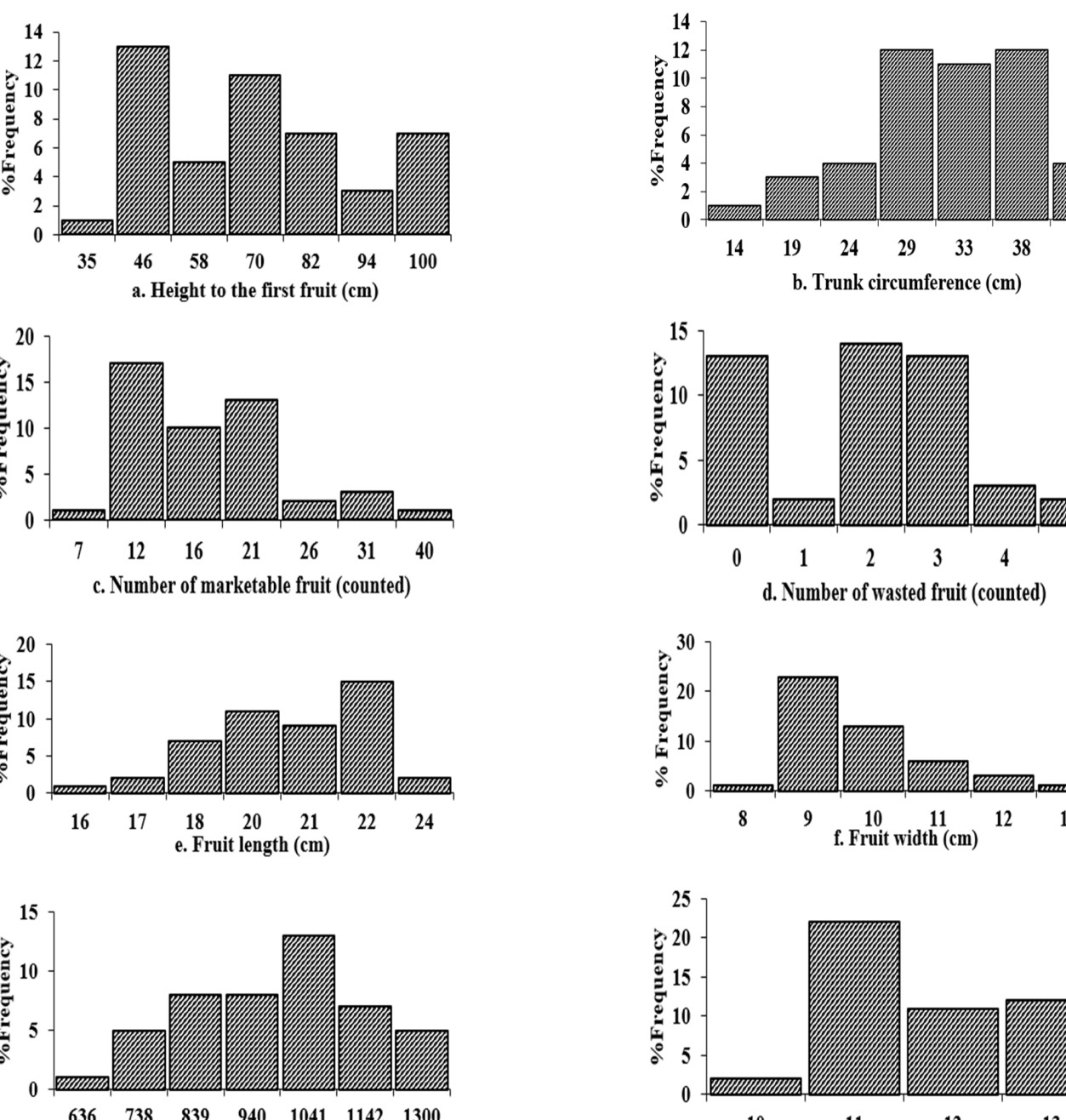

**Figure 1.** *Cont.*

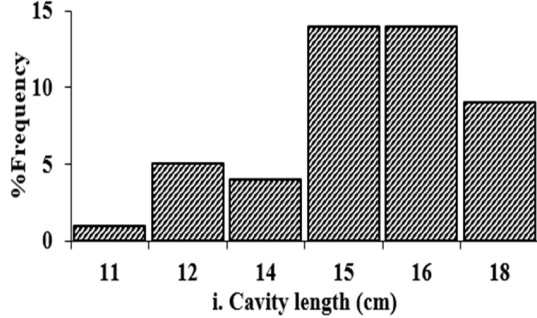

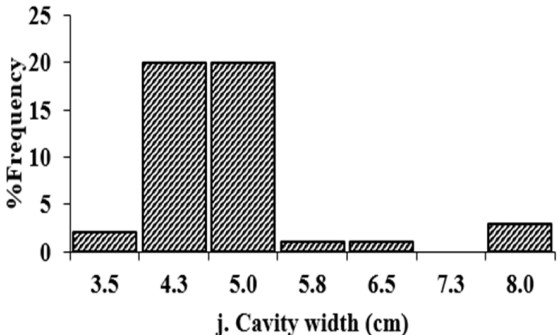

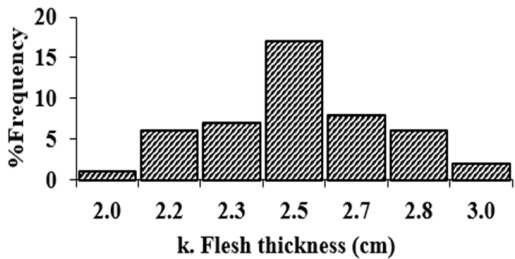

**Figure 1.** (**a–k**). Accumulative phenotype distributions of each trait assessed in the F5 recombinant inbred line population were collected across all four trial sites (*n* = 160).

**Table 2.** Broad sense heritability ($h^2_{b.s}$) of key agronomic and consumer-driven fruit quality traits assessed in the F5 RIL at each environment compared to commercial RB1. Where T1 and T2 are Tablelands trial sites, and C1 and C2 are the Coastal trail sites.

| Traits | $h^2_{b.s}$ at T1 | $h^2_{b.s}$ at T2 | $h^2_{b.s}$ at C1 | $h^2_{b.s}$ at C2 |
|---|---|---|---|---|
| Height to the first fruit (cm) | 0.62 | 0.81 | 0.89 | 0.16 |
| Trunk circumference (cm) | 0.12 | 0.80 | 0.88 | 0.44 |
| Number of marketable fruit (counted) | 0.71 | 0.74 | 0.88 | 0.65 |
| Number of wasted fruit (counted) | 0.86 | 0.10 | 0.13 | 0.82 |
| Fruit weight (g) | 0.89 | 0.18 | 0.48 | 0.00 |
| Fruit width (cm) | 0.90 | 0.73 | 0.44 | 0.15 |
| Fruit length (cm) | 0.90 | 0.11 | 0.16 | 0.29 |
| Flesh thickness (cm) | 0.84 | 0.90 | 0.73 | 0.12 |
| Cavity width (cm) | 0.63 | 0.10 | 0.79 | 0.11 |
| Cavity length (cm) | 0.90 | 0.83 | 0.42 | 0.12 |
| Soluble solid contents (°Brix) | 0.90 | 0.67 | 0.68 | 0.83 |

### 3.3. Breeding Value or Genetic Worth of F5 RIL for Measured Traits

Small variations were detected among the F5 RIL in the Tablelands region for fruit width (9.47–10.49 cm), fruit length (19.78 cm), flesh thickness (2.44 cm), cavity width (4.40–4.45 cm), and cavity length (14.94–15.07 cm). Overall, F5 RIL T1–5-5.9 and T2–5-5.27 performed the best in the Tablelands region, with the first fruit set closest to the ground (T1: 53.49 cm ± 3.36 and T2: 53.46 cm ± 3.32). This was far lower than RB1 (T1: 75.15 cm ± 6.23 and T2: 75.62 cm ± 8.12). In the Tablelands environment, these F5 RIL also had a thicker trunk (T1: 34.66 cm ± 1.69 and T2: 32.15 cm ± 1.69) than RB1 (T1: 25.17 cm ± 1.03 and T2: 26.13 cm ± 1.12), and fewer wasted fruit (T1: 1.53 ± 0.44 and T2: 1.56 ± 0.55) than RB1 (T1: 15.98 ± 0.01 and T2: 13.21 ± 0.05). Moreover, they produced more marketable fruit (T1: 13.73 ± 1.32 and T2: 13.84 ± 1.32) than RB1 (T1: 11.53 ± 1.62 and T2: 11.51 ± 1.51) and produced a more sought-after medium sized (900 g) sweeter fruit (T1: 11.10 ± 0.47 and T2: 11.97 ± 0.47) than the 1300g RB1 (T1: 10.41 ± 0.93 and T2: 10.44 ± 0.95) (Table 4).

**Table 3.** Percentage genetic advances (GA) for each trait assessed in the F5 RIL at each environment at a 10% selection intensity. Where T1 and T2 are Tableland trial sites, and C1 and C2 are the Coastal trial sites.

| Traits | GA at T1 (%) | GA at T2 (%) | GA at C1 (%) | GA at C2 (%) |
|---|---|---|---|---|
| Height to the first fruit (cm) | 16.83 | 17.00 | 5.37 | 14.51 |
| Trunk circumference (cm) | 10.06 | 4.38 | 0.58 | 9.41 |
| Number of marketable fruit (counted) | 0.15 | 0.11 | 0.32 | 0.31 |
| Number of wasted fruit (counted) | 1.15 | 2.31 | 0.15 | 3.18 |
| Fruit weight (g) | 13.38 | 14.76 | 16.94 | 17.77 |
| Fruit width (cm) | 0.72 | 0.91 | 1.09 | 0.76 |
| Fruit length (cm) | 0.58 | 0.13 | 0.09 | 0.58 |
| Flesh thickness (cm) | 0.15 | 0.12 | 0.15 | 0.11 |
| Cavity width (cm) | 0.12 | 0.13 | 3.38 | 0.22 |
| Cavity length (cm) | 0.66 | 0.02 | 0.30 | 0.16 |
| Soluble solid contents (°Brix) | 0.26 | 0.58 | 0.39 | 0.57 |

**Table 4.** F5 RIL ranked for the breeding value or genetic worth (BLUPs) of each trait in the Tablelands environment (T1 and T2) compared to commercial RB1.

| Traits | Line | Location | BLUP | Standard Error (±) |
|---|---|---|---|---|
| Height to the first fruit (cm) | T2-5-5.27 | T2 | 60.55 | 3.36 |
| | T2-5-3.12 | T1 | 59.06 | 3.32 |
| | T2-5-3.12 | T2 | 56.37 | 3.36 |
| | T3-5-6.10 | T1 | 55.96 | 3.36 |
| | T1-5-2.3 | T1 | 55.84 | 3.43 |
| | T1-5-5.9 | T1 | 55.44 | 3.32 |
| | T3-5-6.10 | T2 | 54.87 | 3.60 |
| | T2-5-5.27 | T1 | 54.16 | 3.32 |
| | T1-5-5.9 | T2 | 53.90 | 3.36 |
| | RB1 | T1 | 71.1 | 11.15 |
| | RB1 | T2 | 72.6 | 10.12 |
| Trunk circumference (cm) | T3-5-6.10 | T1 | 31.92 | 1.65 |
| | T3-5-6.10 | T2 | 30.66 | 1.93 |
| | T1-5-2.3 | T1 | 31.96 | 1.65 |
| | T1-5-5.9 | T1 | 30.56 | 1.65 |
| | T1-5-5.9 | T2 | 34.66 | 1.69 |
| | T2-5-5.27 | T1 | 32.15 | 1.65 |
| | T2-5-5.27 | T2 | 29.62 | 1.69 |
| | T2-5-3.12 | T1 | 32.57 | 1.65 |
| | T2-5-3.12 | T2 | 33.19 | 1.69 |
| | RB1 | T1 | 32.11 | 1.03 |
| | RB1 | T2 | 31.55 | 1.12 |
| Number of marketable fruit (counted) | T3-5-6.10 | T1 | 13.44 | 1.33 |
| | T3-5-6.10 | T2 | 11.77 | 1.55 |
| | T1-5-2.3 | T1 | 12.28 | 1.34 |
| | T1-5-5.9 | T1 | 13.64 | 1.32 |
| | T1-5-5.9 | T2 | 13.73 | 1.32 |
| | T2-5-5.27 | T1 | 13.84 | 1.32 |
| | T2-5-5.27 | T2 | 13.89 | 1.32 |
| | T2-5-3.12 | T1 | 10.08 | 1.32 |
| | T2-5-3.12 | T2 | 13.55 | 1.32 |
| | RB1 | T1 | 14.88 | 1.62 |
| | RB1 | T2 | 14.95 | 1.51 |

**Table 4.** *Cont.*

| Traits | Line | Location | BLUP | Standard Error (±) |
|---|---|---|---|---|
| Number of Wasted fruit (counted) | T3-5-6.10 | T1 | 2.64 | 0.44 |
| | T3-5-6.10 | T2 | 1.72 | 0.59 |
| | T1-5-2.3 | T1 | 1.04 | 0.44 |
| | T1-5-5.9 | T1 | 1.38 | 0.44 |
| | T1-5-5.9 | T2 | 1.53 | 0.45 |
| | T2-5-5.27 | T1 | 1.56 | 0.44 |
| | T2-5-5.27 | T2 | 1.53 | 0.45 |
| | T2-5-3.12 | T1 | 1.52 | 0.44 |
| | T2-5-3.12 | T2 | 1.79 | 0.45 |
| | RB1 | T1 | 1.98 | 0.01 |
| | RB1 | T2 | 2.11 | 0.05 |
| Fruit weight (g) | T3-5-6.10 | T1 | 958.13 | 0.73 |
| | T3-5-6.10 | T2 | 957.67 | 0.11 |
| | T1-5-2.3 | T1 | 916.95 | 0.18 |
| | T1-5-5.9 | T1 | 1016.39 | 0.14 |
| | T1-5-5.9 | T2 | 937.04 | 0.91 |
| | T2-5-5.27 | T1 | 879.92 | 1.11 |
| | T2-5-5.27 | T2 | 989.99 | 0.88 |
| | T2-5-3.12 | T1 | 935.07 | 0.51 |
| | T2-5-3.12 | T2 | 982.06 | 0.42 |
| | RB1 | T1 | 1026.67 | 75.51 |
| | RB1 | T2 | 1028.11 | 72.23 |
| Fruit width (cm) | T3-5-6.10 | T1 | 9.87 | 0.34 |
| | T3-5-6.10 | T2 | 9.95 | 0.38 |
| | T1-5-2.3 | T1 | 9.73 | 0.36 |
| | T1-5-5.9 | T1 | 10.38 | 0.34 |
| | T1-5-5.9 | T2 | 9.79 | 0.34 |
| | T2-5-5.27 | T1 | 9.54 | 0.34 |
| | T2-5-5.27 | T2 | 9.73 | 0.34 |
| | T2-5-3.12 | T1 | 9.66 | 0.34 |
| | T2-5-3.12 | T2 | 10.49 | 0.34 |
| | RB1 | T1 | 11.36 | 1.45 |
| | RB1 | T2 | 12.11 | 1.56 |
| Flesh thickness (cm) | T3-5-6.10 | T1 | 2.446 | 0.01 |
| | T3-5-6.10 | T2 | 2.446 | 0.01 |
| | T1-5-2.3 | T1 | 2.446 | 0.01 |
| | T1-5-5.9 | T1 | 2.446 | 0.01 |
| | T1-5-5.9 | T2 | 2.446 | 0.01 |
| | T2-5-5.27 | T1 | 2.446 | 0.01 |
| | T2-5-5.27 | T2 | 2.446 | 0.01 |
| | T2-5-3.12 | T1 | 2.446 | 0.01 |
| | T2-5-3.12 | T2 | 2.446 | 0.01 |
| | RB1 | T1 | 2.92 | 0.41 |
| | RB1 | T2 | 2.94 | 0.45 |
| Cavity width (cm) | T3-5-6.10 | T1 | 4.55 | 0.15 |
| | T3-5-6.10 | T2 | 4.69 | 0.18 |
| | T1-5-2.3 | T1 | 4.56 | 0.15 |
| | T1-5-5.9 | T1 | 4.72 | 0.15 |
| | T1-5-5.9 | T2 | 4.40 | 0.15 |
| | T2-5-5.27 | T1 | 4.34 | 0.15 |
| | T2-5-5.27 | T2 | 4.67 | 0.15 |
| | T2-5-3.12 | T1 | 4.55 | 0.01 |
| | T2-5-3.12 | T2 | 4.73 | 0.01 |
| | RB1 | T1 | 5.81 | 1.11 |
| | RB1 | T2 | 5.98 | 1.52 |

**Table 4.** *Cont.*

| Traits | Line | Location | BLUP | Standard Error (±) |
|---|---|---|---|---|
| Fruit length (cm) | T3-5-6.10 | T1 | 19.78 | 0.01 |
| | T3-5-6.10 | T2 | 19.78 | 0.01 |
| | T1-5-2.3 | T1 | 19.78 | 0.01 |
| | T1-5-5.9 | T1 | 19.78 | 0.01 |
| | T1-5-5.9 | T2 | 19.78 | 0.01 |
| | T2-5-5.27 | T1 | 19.78 | 0.01 |
| | T2-5-5.27 | T2 | 19.78 | 0.01 |
| | T2-5-3.12 | T1 | 19.78 | 0.01 |
| | T2-5-3.12 | T1 | 19.78 | 0.01 |
| | RB1 | T1 | 21.32 | 1.51 |
| | RB1 | T2 | 21.55 | 1.52 |
| Cavity length (cm) | T3-5-6.10 | T1 | 14.94 | 0.46 |
| | T3-5-6.10 | T2 | 15.43 | 0.47 |
| | T1-5-2.3 | T1 | 14.96 | 0.49 |
| | T1-5-5.9 | T1 | 15.07 | 0.45 |
| | T1-5-5.9 | T2 | 15.39 | 0.45 |
| | T2-5-5.27 | T1 | 15.33 | 0.45 |
| | T2-5-5.27 | T2 | 14.90 | 0.45 |
| | T2-5-3.12 | T1 | 15.37 | 0.45 |
| | T2-5-3.12 | T2 | 15.04 | 0.45 |
| | RB1 | T1 | 16.45 | 1.31 |
| | RB1 | T2 | 16.99 | 1.22 |
| Soluble solid contents (°Brix) | T3-5-6.10 | T1 | 11.95 | 0.47 |
| | T3-5-6.10 | T2 | 11.06 | 0.51 |
| | T1-5-2.3 | T1 | 11.84 | 0.51 |
| | T1-5-5.9 | T1 | 10.85 | 0.47 |
| | T1-5-5.9 | T2 | 11.10 | 0.47 |
| | T2-5-5.27 | T1 | 11.97 | 0.47 |
| | T2-5-5.27 | T2 | 11.88 | 0.47 |
| | T2-5-3.12 | T1 | 11.89 | 0.47 |
| | T2-5-3.12 | T2 | 11.14 | 0.47 |
| | RB1 | T1 | 10.41 | 0.93 |
| | RB1 | T2 | 10.44 | 0.95 |

In the Coastal region, the F5 RIL performed similarly for fruit weight (959 g), fruit length (19.15–19.78 cm), fruit width (9.54 cm) and flesh thickness (2.35–2.46 cm) to those grown on the Tablelands. The F5 RIL C3-3-5.24 and C2-5-5 had the highest genetic worth, with the first fruit set closer to the ground (C1: 63.69 cm ± 6.68 and C2: 81.81 cm ± 7.18) than RB1 (C1: 110.92 cm ± 7.89 and C2: 101.23 cm ± 8.12), a thicker trunk circumference (C1: 31.82 cm ± 2.88 and C2: 31.61 cm ± 2.89) than RB1 (C1: 31.10 cm ± 2.85 and C2: 31.25 ± 2.92), and fewer wasted fruit (C1: 0.55 ± 0.44 and C2: 0.67 ± 0.47) than RB1 (C1: 2.51 ± 0.43 and C2: 2.55 ± 0.49) (Table 5 ). Moreover, these F5 RIL produced more marketable fruit (C1: 19.03 ± 1.62 and C2: 19.06 ± 1.63) than RB1 (C1: 14.21 ± 2.34 and C2: 15.23 ± 2.45) and of a medium size (around 900 g) that was sweeter (C1: 11.87 ± 0.30 and T2: 11.96 ± 0.30) than RB1 (C1: 9.49 ± 0.52 and C2: 9.52 ± 0.31) (Table 5).

**Table 5.** F5 RIL ranked for the breeding value or genetic worth (BLUPs) of each trait in the Coastal environment (C1 and C2) compared to commercial RB1.

| Traits | Line | Location | BLUP | Standard Error (±) |
|---|---|---|---|---|
| Height to the first fruit (cm) | C1-5-4.1 | C1 | 91.15 | 6.68 |
| | C1-5-4.1 | C2 | 75.84 | 7.11 |
| | C1-5-4.2 | C1 | 89.92 | 6.76 |
| | C1-5-4.3 | C1 | 82.70 | 6.76 |
| | C3-3-5.24 | C1 | 63.69 | 6.68 |
| | C3-3-5.24 | C2 | 64.94 | 7.11 |
| | C2-5-5 | C2 | 81.81 | 7.18 |
| | RB1 | C1 | 100.92 | 7.89 |
| | RB1 | C2 | 101.23 | 8.12 |

**Table 5.** *Cont.*

| Traits | Line | Location | BLUP | Standard Error (±) |
|---|---|---|---|---|
| Trunk circumference (cm) | C1-5-4.1 | C1 | 31.59 | 2.68 |
| | C1-5-4.1 | C2 | 27.68 | 2.88 |
| | C1-5-4.2 | C1 | 33.05 | 2.69 |
| | C1-5-4.3 | C1 | 28.89 | 2.69 |
| | C3-3-5.24 | C1 | 20.92 | 2.68 |
| | C3-3-5.24 | C2 | 31.82 | 2.88 |
| | C2-5-5 | C2 | 31.61 | 2.89 |
| | RB1 | C1 | 31.10 | 1.85 |
| | RB1 | C2 | 31.25 | 1.92 |
| Number of marketable fruit (counted) | C1-5-4.1 | C1 | 19.06 | 1.62 |
| | C1-5-4.1 | C2 | 20.83 | 1.62 |
| | C1-5-4.2 | C1 | 20.05 | 1.66 |
| | C1-5-4.3 | C1 | 19.62 | 1.66 |
| | C3-3-5.24 | C1 | 19.03 | 1.62 |
| | C3-3-5.24 | C2 | 19.35 | 1.62 |
| | C2-5-5 | C1 | 19.06 | 1.62 |
| | RB1 | C1 | 14.21 | 2.34 |
| | RB1 | C2 | 15.23 | 2.45 |
| Number of Wasted fruit (counted) | C1-5-4.1 | C1 | 0.84 | 0.44 |
| | C1-5-4.1 | C2 | 1.81 | 0.45 |
| | C1-5-4.2 | C1 | 0.85 | 0.46 |
| | C1-5-4.3 | C1 | 0.85 | 0.46 |
| | C3-3-5.24 | C1 | 1.01 | 0.44 |
| | C3-3-5.24 | C2 | 0.55 | 0.45 |
| | C2-5-5 | C2 | 0.67 | 0.47 |
| | RB1 | C1 | 1.51 | 0.43 |
| | RB1 | C2 | 1.55 | 0.49 |
| Fruit weight (g) | C1-5-4.1 | C1 | 959.46 | 0.01 |
| | C1-5-4.1 | C2 | 959.46 | 0.01 |
| | C1-5-4.2 | C1 | 959.46 | 0.01 |
| | C1-5-4.3 | C1 | 959.46 | 0.01 |
| | C3-3-5.24 | C1 | 959.46 | 0.01 |
| | C3-3-5.24 | C2 | 959.46 | 0.01 |
| | C2-5-5 | C2 | 959.46 | 0.01 |
| | RB1 | C1 | 1056.45 | 84.42 |
| | RB1 | C2 | 1102.56 | 79.23 |
| Fruit width (cm) | C1-5-4.1 | C1 | 9.54 | 0.01 |
| | C1-5-4.1 | C2 | 9.54 | 0.01 |
| | C1-5-4.2 | C1 | 9.54 | 0.01 |
| | C1-5-4.3 | C1 | 9.54 | 0.01 |
| | C3-3-5.24 | C1 | 9.54 | 0.01 |
| | C3-3-5.24 | C2 | 9.54 | 0.01 |
| | C2-5-5 | C2 | 9.54 | 0.01 |
| | RB1 | C1 | 9.56 | 0.45 |
| | RB1 | C2 | 9.61 | 0.61 |
| Flesh thickness (cm) | C1-5-4.1 | C1 | 2.40 | 0.11 |
| | C1-5-4.1 | C2 | 2.46 | 0.11 |
| | C1-5-4.2 | C1 | 2.35 | 0.11 |
| | C1-5-4.3 | C1 | 2.32 | 0.11 |
| | C3-3-5.24 | C1 | 2.74 | 0.11 |
| | C3-3-5.24 | C2 | 2.36 | 0.11 |
| | C2-5-5 | C2 | 2.35 | 0.11 |
| | RB1 | C1 | 2.33 | 0.26 |
| | RB1 | C2 | 2.31 | 0.23 |

**Table 5.** *Cont.*

| Traits | Line | Location | BLUP | Standard Error (±) |
|---|---|---|---|---|
| Cavity width (cm) | C1-5-4.1 | C1 | 4.38 | 0.56 |
| | C1-5-4.1 | C2 | 4.67 | 0.58 |
| | C1-5-4.2 | C1 | 3.91 | 0.56 |
| | C1-5-4.3 | C1 | 5.61 | 0.56 |
| | C3-3-5.24 | C1 | 6.39 | 0.56 |
| | C3-3-5.24 | C2 | 4.67 | 0.58 |
| | C2-5-5 | C2 | 4.41 | 0.58 |
| | RB1 | C1 | 6.86 | 0.49 |
| | RB1 | C2 | 6.89 | 4.51 |
| Fruit length (cm) | C1-5-4.1 | C1 | 19.78 | 0.43 |
| | C1-5-4.1 | C2 | 19.54 | 0.43 |
| | C1-5-4.2 | C1 | 19.50 | 0.44 |
| | C1-5-4.3 | C1 | 19.63 | 0.44 |
| | C3-3-5.24 | C1 | 19.18 | 0.43 |
| | C3-3-5.24 | C2 | 19.81 | 0.43 |
| | C2-5-5 | C2 | 19.15 | 0.44 |
| | RB1 | C1 | 19.41 | 0.31 |
| | RB1 | C2 | 19.52 | 0.30 |
| Cavity length (cm) | C1-5-4.1 | C1 | 15.13 | 0.51 |
| | C1-5-4.1 | C2 | 14.61 | 0.52 |
| | C1-5-4.2 | C1 | 14.39 | 0.53 |
| | C1-5-4.3 | C1 | 14.99 | 0.53 |
| | C3-3-5.24 | C1 | 13.65 | 0.51 |
| | C3-3-5.24 | C2 | 15.28 | 0.52 |
| | C2-5-5 | C2 | 14.16 | 0.53 |
| | RB1 | C1 | 16.96 | 0.44 |
| | RB1 | C2 | 16.45 | 0.41 |
| Soluble solid contents (°Brix) | C1-5-4.1 | C1 | 10.92 | 0.29 |
| | C1-5-4.1 | C2 | 11.68 | 0.30 |
| | C1-5-4.2 | C1 | 11.81 | 0.29 |
| | C1-5-4.3 | C1 | 11.05 | 0.29 |
| | C3-3-5.24 | C1 | 11.70 | 0.29 |
| | C3-3-5.24 | C2 | 10.87 | 0.30 |
| | C2-5-5 | C2 | 10.96 | 0.30 |
| | RB1 | C1 | 9.49 | 0.25 |
| | RB1 | C2 | 9.52 | 0.31 |

*3.4. F5 RIL Trait Gain Percentage (Increase or Decrease) over RB1*

In the Tablelands region, RIL T1-5-5.9 and T2-5-5.27 exhibited the greatest average trait gains, including up to a 54% reduction in height to first fruit, and produced 18% more marketable fruit than RB1. They also produced fruit with a 6% average smaller cavity that were up to 36% smaller and 21% sweeter than RB1. Additionally, they had a 37% larger average trunk circumference than RB1 (Table 6).

In the Coastal region, RIL C3-3-5.24 and C2-5-5 exhibited the greatest average trait gains, including up to a 32% reduction in height to first fruit, and produced 53% more marketable fruit than RB1. They also produced fruit that were up to 7% smaller, had a 6% reduced fruit cavity and were up to 21% sweeter than RB1. Additionally, they had a 29% larger trunk circumference than RB1 (Table 5).

**Table 6.** Trait gain percentage (%) increased or decreased in each particular trait within each F5 RIL in each environment either (T1 + T2) or (C1 + C2) compared to commercial RB1.

| Traits | T3-5-6.10 | T1-5-2.3 | T1-5-5.9 | T2-5-5.27 | T2-5-3.12 | C1-5-4.1 | C1-5-4.1 | C1-5-4.3 | C3-3-5.24 | C2-5-5 |
|---|---|---|---|---|---|---|---|---|---|---|
| Height to the first fruit (cm) | −49.2 | −47.6 | −53.0 | −54.6 | −12.7 | −0.48 | −1.13 | −9.81 | −32.96 | −30.71 |
| Trunk circumference (cm) | −29.7 | −29.4 | 36.5 | 28.5 | 7.0 | −38.06 | −33.96 | −44.78 | 29.93 | 22.39 |
| Number of Marketable fruit (counted) | 18.4 | −0.2 | −13.9 | 21.6 | 14.8 | 53.52 | 52.20 | 40.85 | 50.01 | 53.52 |
| Number of Wasted fruit (counted) | −100 | −33.3 | −100 | −100 | 11.1 | 100.00 | −100.00 | −100.00 | −100.00 | −100.00 |
| Fruit weight (g) | −13.7 | −33.2 | −6.4 | −39.4 | −9.3 | 12.56 | −16.18 | 9.79 | −7.18 | −3.20 |
| Fruit width (cm) | −16.9 | −19.8 | −8.4 | −24.4 | −3.3 | 3.16 | −4.91 | 18.25 | 8.77 | −11.40 |
| Fruit length (cm) | 3.5 | −6.7 | 13.5 | −5.5 | 1.8 | 8.70 | 5.80 | 4.35 | 27.54 | −3.62 |
| Flesh thickness (cm) | −28.6 | −28.1 | −22.6 | −35.7 | −20.9 | −6.21 | −18.62 | 26.21 | 46.90 | −17.24 |
| Cavity width (cm) | 1.8 | −6.6 | 6.3 | −6.2 | 7.4 | 6.55 | 0.69 | 3.28 | −4.31 | −6.47 |
| Cavity length (cm) | −2.6 | −21.6 | 6.0 | −7.8 | 15.0 | 17.22 | 7.59 | 15.44 | 0.25 | 5.70 |
| Soluble solid contents (°Brix) | 1.17 | 7.8 | 20.56 | 21.64 | 4.9 | 18.15 | 9.89 | 9.93 | 21.47 | 13.70 |

## 4. Discussion

The advanced breeding lines assessed in this study were significantly improved for several key agronomic and consumer-driven fruit quality traits over the current standard commercial red papaya variety, RB1. This included lines that were stable for traits such as fruit size, with fruit of different sizes providing opportunity for defined domestic and international markets [25]. Additionally, substantial reductions were achieved in fruit cavity size, reflecting an increase in the amount of edible flesh, weight, and hence potentially increased economic fruit value.

Cavity width and wasted fruit traits were found to be highly heritable with a proposed monogenic or simple inheritance prediction. Additionally, a significant genetic gain was already made for these traits through breeding compared to RB1 with some variation remaining among the F5 RIL. This provides an excellent chance to continue to improve these traits through simple selective breeding approaches. Meanwhile, several other traits were not found to be highly heritable in this study, particularly at some field sites, including trunk circumference at T1, fruit length at T2, number of wasted fruit at T2 and C1, and height to the first, fruit weight, fruit width, flesh thickness, and cavity length at C2. These traits were also proposed to be digenic or polygenic in nature, indicating that they are underpinned by far more complex genetics. This was the case for traits; plant yield, height to the first fruit, trunk circumference, fruit weight, fruit area (length and width), and soluble solid contents (°Brix) with low heritability (<0.2) in papaya [15,16,26].

In addition to underlying genetics, the environment is crucial in inducing genetic alterations in key agronomic productivity and fruit quality traits [27–29]. However, within the RIL in the current study, the expression of some of the target traits significantly varied between the two growing regions. Since there are clear differences in the climates among these regions, the variations observed are likely due to climatic effects, such as spatial and volumetric differences in rainfall and mean temperatures. Similarly, major environmental influences were previously demonstrated to affect the expression of fruit size (length and width), fruit weight, and soluble solid contents (°Brix) in apple (*Malus pomila*) [11], citrus (*Citrus* spp.) [12], and avocado (*Persea americana*) [13].

Other environmental factors are also likely to have influenced trait expression [10,30] within F5 RIL at a single field site or among field sites within the same region, including clay to sandy loam in soil type, uneven distributions of fertilizers (broadcasting to drip fertigation), planting method (double row or single row) and planting time, and tree spatial distribution (tree-to-tree and row-to-row spacing). These on-farm factors and cultural practices have previously been determined to significantly contribute to variation in fruit yield, fruit weight, and soluble solid content (°Brix) expression in lemon (*Citrus lemon*) [31] and citrus (*Citrus* spp.) [32]. The soil effect was evident in contributing to variation in plant growth and production patterns and carbon partitioning [33] between roots and shoots in avocado (*Persea americana* Mill) [34]. A deeper assessment of microclimate and on-farm environmental and farming practices on a larger sample population at each of the trial



sites would be required to tease apart the finer non-genetic effects on the papaya trait expressions. This would better inform optimal tree growing conditions for the highest productivity and fruit quality. Specifically, land preparation, planting, water inputs, and nutrient management practices must be standardized to achieve maximum yields and net benefits across different growing locations [35–37].

## 5. Conclusions

The genetic worth of each RIL for key stabilized traits was measured and agreed with the findings of [22] that provided a baseline for selecting the best performing individuals for Tablelands and Coastal regions. The results obtained were used for selecting lines with superior performance for key productive and fruit quality traits. The best performing individuals (T3-5-6.10 and T1-5-2.3 for Tablelands) and (C3-3-5.24 and C2-5-5 for Coast) were selected for advancement to F7 and the production of F1 hybrid cultivars that suit industry requirements and consumer needs.

**Author Contributions:** Conceptualization, F.A.; methodology, F.A.; software, F.A.; validation, F.A., C.K.-u. and R.F.; formal analysis, F.A.; investigation, F.A., C.K.-u. and R.F.; resources, F.A., C.K.-u. and R.F.; data curation, F.A.; writing—original draft preparation, F.A.; writing—review and editing, F.A., C.K.-u. and R.F.; visualization, F.A., C.K.-u. and R.F.; supervision, R.F.; project administration, F.A., C.K.-u. and R.F.; funding acquisition, R.F. All authors have read and agreed to the published version of the manuscript.

**Funding:** The current study results are a part of the National Papaya Breeding and Evaluation Program (PP18000)-National Papaya Breeding and Evaluation Program has been funded by Hort Innovation, using the papaya industry research and development levy and contributions from the Australian Government. Hort Innovation is the grower-owned, not-for-profit research and development corporation for Australian horticulture.

**Institutional Review Board Statement:** No applicable.

**Informed Consent Statement:** No applicable.

**Data Availability Statement:** No applicable.

**Conflicts of Interest:** The authors declare no conflict of interest.

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
