# Peer review of "The Inheritance Pattern of Key Desirable Agronomic and Fruit Quality Traits in Elite Red Papaya Genotypes"

_horticulturae, doi:10.3390/horticulturae8090845_

Round 1

Reviewer 1 Report

The authors do an interesting work on Carica papaya L. should be expanded by giving information on the four quadrants that they establish in the tree. To this end, they should indicate to which geographical orientation each of these quadrants corresponds. 

It is known from previous studies that this clearly influences the development and growth of the fruit.

It is necessary to carry out sensory evaluations on the varieties made to complete the study of new varieties.

Author Response

Reviewer 1

Comments and Suggestions for Authors

The authors do an interesting work on Carica papaya L. should be expanded by giving information on the four quadrants that they establish in the tree. To this end, they should indicate to which geographical orientation each of these quadrants corresponds. It is known from previous studies that this clearly influences the development and growth of the fruit.

Author’s response: The authors agreed to provide more information about each quadrant selected in the Methods section. The quadrant represents each side of the papaya tree from the centre of the crown.

Line 123-124: Authors rephrased the sentence “Subsequently, the entire fruit column of each tree (replicate) was visually divided into four quadrants (representing each side of the tree from the centre of the crown)”

It is necessary to carry out sensory evaluations on the varieties made to complete the study of new varieties.

Author’s Response: This is not in the scope of the present study; however, the authors appreciate the suggestion provided by the reviewer and would like to include it in future studies (focusing on sensory evaluations of new red papaya genotypes).

Reviewer 2 Report

Comments and Suggestions for Authors

Manuscript ID: horticulturae-1883377

Title: The inheritance pattern of key desirable agronomic and fruit quality traits in elite red papaya genotypes.

 In my opinion this manuscript should be: Reconsider after major revision 

The authors need address many aspects in order to the manuscript can be accepted in this journal:

For the entire manuscript:

In Instructions for Authors, it specifies “References must be numbered in order of appearance in the text (including table captions and figure legends) and listed individually at the end of the manuscript. In the text, reference numbers should be placed in square brackets [], and placed before the punctuation”. But throughout this manuscript the references are not written as indicated in the instructions.

Abstract: 

In Instructions for Authors, it specifies “the abstract should be a total of about 200 words maximum”. This abstract has 360 words. It must be shortened.

Introduction:

1.       In lines 42, 44 and 46, a reference appears (Hort Innovation, 2021), which is not in the list of references.

2.       In line 49, there is reference “hdoa.hawaii.gov”, but not in the list of references.

3.       Also in line 59 there is the reference Visscher et al., 2008, which is not in the list of references.

4.       In lines 69-70 authors wrote “However, no prior reports exist on potential environmental impacts on trait stability or heritability of other important papaya traits”. A quick search allowed me to find several papers (I give three references as an example) in which heritability studies of different characters of papaya are described, as well as genetic correlations between them. In these articles, some of the characters addressed in this study were studied, such as: stem diameter and insertion height of the first fruit. References:

a.       Kumar, M., Prasad, Y., Kumar, M., Prakash, S., & Kumar, S. (2015). Evaluation of Genetic Variability, Genetic advance, Heritability and Character association for Yield and its Contributing traits in Papaya (Carica papaya L.). Society Plant Research, 28(2), 99-102.

b.       Pereira, M. G., Ramos, H. C. C., Junior, P. C. D., Pereira, T. N. S., & Ide, C. D. (2007). Genotypic correlations of morpho-agronomic traits in papaya and implications for genetic breeding. Crop Breeding and Applied Biotechnology, 7(4).

c.       Karunakaran, G., Ravishankar, H., & Dinesh, M. R. (2008, December). Genetical studies in papaya (Carica papaya L.). In II International Symposium on Papaya 851 (pp. 103-108).

5.       In lines 80-82 authors state “Additionally, the important traits of number of marketable and wasted fruit per tree fruit column were assessed, with the goal of understanding the potential to improve marketable fruit yield”. Please, I would like to know what the authors mean by marketable and wasted fruits. They need to explain what genetic aspects or characteristics they consider for a fruit to be marketable or discarded. The yield of any crop has different components that are genetically determined (for example: fruit size, fruit weight, fruit form, peel thickness, disease resistance, etc.) and their expression is influenced by the environmental conditions in which the plant growth, and many other biotic and abiotic factors. A fruit is wasted or not, as a consequence of the above, and therefore is subject to many variations. In my opinion, wasted fruits is not a genetic trait that we can be improved through crosses or selection. Which can to be improved are the components of the yield, so that there will be fewer fruits wasted. In addition, this “trait” are not part of the list of descriptors that have been proposed for papaya (IPGRI 1988).

Methods:

1.       In line 116, reference is made to Kanchana-udomkan et al., 2018, but in the list of references there are two with the same author and year, it is necessary to distinguish which one it corresponds to.

2.       In this section, in lines 123 to 124, authors describe the four productivity traits, among which is the trait named “number of wasted fruit (counted per fruit column)”. As I wrote above, there are many biotic and abiotic factors that can contribute in different ways and magnitude to a fruit being marketable or to its deterioration and, therefore, to its wastage. In my opinion, the authors need to explain what they were referring to, when they decided to include this measurement.

3.       In line 130 reference is made to Kanchana-udomkan et al., 2020, but in the list of references there is not this author and year.

Results:

1.       In lines 225 to 247 are listed the values obtained for the different characters measured in the different environments, but this information is also found in tables 4a and 4b. Why repeat the information? Only those results that were most genetically significant within the study should be emphasized.

2.       In table 4a, there are very high standard error values, specifically in the trait “Fruit weight (g)” (from ± 59.72 to ± 75.5) it indicates a greater dispersion of the data, and less precise confidence intervals.

3.       It is not possible to see the whole table 5, its dimensions exceed the margins of the manuscript.

Discussion:

1.       In lines 295-299 authors stated “Similarly, major environmental influences were previously demonstrated to affect the expression of fruit size (length and width), fruit weight and soluble solid contents (ºBrix) in apple (Malus pomila; Kumar et al., 2012), citrus (Citrus spp.; Imai et al., 2018) and avocado (Persea americana; Cañas-Gutiérrez et al., 2022)”.  I wonder if the high standard errors presented in table 4, for the case of Fruit weight can be explained only by the influence of the environment. Why, if there are several genetic studies carried out in different countries on papaya, on most of the characters studied here, do the authors refer to studies carried out on other crops, quite taxonomically distant from papaya, even if they are fruit trees?

2.       Line 318, reference Zulfi and Luo (2018) there is not in reference list.

3.       In general, the findings of this study are compared and discussed with the results obtained in Citrus and avocado, which are physiologically different plants from papaya and have longer crop cycles, and different agronomic management. If there were no studies of this type in papaya, the use of these references could be partly explained, but there are several previous studies in India and Brazil on the heritability and expression of many of the characters studied in this research.

References:

1.       The reference that appears with the number 9, Cañas-Gutierrez, this surname is misspelled in the list of references.

2.       In general, there are references in the text that are not in the list of references at the end of the manuscript. And in the list of references there are 18 citations that are not in the text, these are the ones corresponding to the numbers: 3, 5, 6, 10, 15, 17, 18, 22, 24, 26, 35, 36, 37, 38, 39, 40, 41, 43.

Author Response

Reviwer 2

Comments and Suggestions for Authors

Manuscript ID: horticulturae-1883377

Title: The inheritance pattern of key desirable agronomic and fruit quality traits in elite red papaya genotypes.

 In my opinion this manuscript should be: Reconsider after major revision 

The authors need address many aspects in order to the manuscript can be accepted in this journal:

For the entire manuscript:

In Instructions for Authors, it specifies “References must be numbered in order of appearance in the text (including table captions and figure legends) and listed individually at the end of the manuscript. In the text, reference numbers should be placed in square brackets [], and placed before the punctuation”. But throughout this manuscript the references are not written as indicated in the instructions.

Author’s response: The authors agreed to correct the Reference style according to the journal format.

Abstract: 

In Instructions for Authors, it specifies “the abstract should be a total of about 200 words maximum”. This abstract has 360 words. It must be shortened.

Author’s Response:

The authors agreed to shorten the Abstract (please see below).

Abstract

The knowledge of the heritability, genetic advance and stability of key traits, such as the height to the first fruit, trunk circumference, number of marketable fruit, wasted fruit, fruit weight, fruit width, fruit length, flesh thickness, cavity width, cavity length and soluble solid contents is required. These were determined in ten advanced generation red papaya recombinant inbred lines (RIL; F5 generation). The F5 RIL were grown in four field sites, two each within two distinct agroecological climates: the Tablelands and Coastal regions. At each site, biological replicates (trees) of each RIL and the industry-standard red papaya cultivar, RB1, were grown. Agronomic traits and fruit-specific traits were assessed at five and 10 months, respectively, after seedling transplantation to the field. Height to first fruit, trunk circumference, fruit weight and soluble solid contents were highly heritable and stable at all field sites (h2b.s, 0.7-0.9) with genetic gains of almost 18% observed for height to first fruit and fruit weight. Across all sites, the trunks of the F5 lines were 37% wider, the trees set fruit 47% closer to the ground and had 20% more marketable fruit with 33% smaller fruit cavity widths, and their fruit was 11% heavier and 12% sweeter than RB1.

Introduction:

  1. In lines 42, 44 and 46, a reference appears (Hort Innovation, 2021), which is not in the list of references.

Author’s response: The authors reported the reference (Hort Innovation, 2021)

On-Line 42-44: as [7], (p. 173)  

On-Line 44-46: as [7], (p. 171)

On-Line 46-48: as [7], (p. 175)

  1. In line 49, there is the reference “hdoa.hawaii.gov”, but not in the list of references.

Author’s response: The authors reported the reference (hdoa.hawaii.gov)

On-Line 46-48: as [38], (p. 1)

  1. Also in line 59 there is the reference Visscher et al., 2008, which is not in the list of references.

Author’s response: The authors reported the reference [1,2,11,16]

  1. In lines 69-70 authors wrote “However, no prior reports exist on potential environmental impacts on trait stability or heritability of other important papaya traits”. A quick search allowed me to find several papers (I give three references as an example) in which heritability studies of different characters of papaya are described, as well as genetic correlations between them. In these articles, some of the characters addressed in this study were studied, such as: stem diameter and insertion height of the first fruit. References:
    1. Kumar, M., Prasad, Y., Kumar, M., Prakash, S., & Kumar, S. (2015). Evaluation of Genetic Variability, Genetic advance, Heritability and Character association for Yield and its Contributing traits in Papaya (Carica papaya). Society Plant Research, 28(2), 99-102.
    2. Pereira, M. G., Ramos, H. C. C., Junior, P. C. D., Pereira, T. N. S., & Ide, C. D. (2007). Genotypic correlations of morpho-agronomic traits in papaya and implications for genetic breeding. Crop Breeding and Applied Biotechnology, 7(4).
    3. Karunakaran, G., Ravishankar, H., & Dinesh, M. R. (2008, December). Genetical studies in papaya (Carica papaya). In II International Symposium on Papaya 851(pp. 103-108).

Author’s Response: The authors described the potential environmental impacts of two distinct agro-geographical climates each with two trial sites on papaya tree productivity and fruit quality traits in F5 RIL. However, the given references by the reviewer are confined to a single environment study (single location) and in retrospect trait stability single environment study is not sufficient.

Therefore, the authors reported in lines 69-70 “However, no prior reports exist on potential environmental impacts on trait stability or heritability of other important papaya traits”.

In lines 80-82 authors state “Additionally, the important traits of number of marketable and wasted fruit per tree fruit column were assessed, with the goal of understanding the potential to improve marketable fruit yield”. Please, I would like to know what the authors mean by marketable and wasted fruits. They need to explain what genetic aspects or characteristics they consider for a fruit to be marketable or discarded. The yield of any crop has different components that are genetically determined (for example: fruit size, fruit weight, fruit form, peel thickness, disease resistance, etc.) and their expression is influenced by the environmental conditions in which the plant growth, and many other biotic and abiotic factors. A fruit is wasted or not, as a consequence of the above, and therefore is subject to many variations. In my opinion, wasted fruits is not a genetic trait that we can be improved through crosses or selection. Which can to be improved are the components of the yield, so that there will be fewer fruits wasted. In addition, this “trait” are not part of the list of descriptors that have been proposed for papaya (IPGRI 1988).

Author’s Response: The authors would like to provide a clear demonstration of “wasted Fruit”. The wasted fruit is a trait that we measured by counting the number of carpelloid (partially developed) fruit on the fruit column at a similar time when we actually counted the number of marketable (fully developed) fruit. Wasted fruit is not a dropped fruit on the ground it is actual fruit but partially developed (mostly due to poor pollination) on the fruit column and is not saleable in the market from the farm gate. Therefore, any partially developed fruit developed on each fruit column was referred to as “Wasted Fruit (counted per fruit column).

The authors showed Wasted fruit is a loss to the productivity trait (marketable fruit) and is under bimodal digenic inheritance as shown in Figure 1d. The authors believe in the considerable effect of the environment (biotic and abiotic) on the production of “wasted fruit” per fruit column in papaya. The authors included the trait “wasted fruit” in the breeding strategy during the selection of the most promising breeding lines as a key trait to reduce the number of wasted fruit produced on each fruit column across the genotypes. Thus, if wasted fruit is produced less on each fruit column that will entail enhancing the production of marketable fruit.

The authors would like to propose adding “Wasted fruit” as a trait in the revised version of (IPGRI 1988).

Methods:

  1. In line 116, reference is made to Kanchana-udomkan et al., 2018, but in the list of references there are two with the same author and year, it is necessary to distinguish which one it corresponds to.

Author’s Response: In Line 116 the reference (Kanchana-udomkan et al., 2018) has been added as [21].

  1. In this section, in lines 123 to 124, authors describe the four productivity traits, among which is the trait named “number of wasted fruit (counted per fruit column)”. As I wrote above, there are many biotic and abiotic factors that can contribute in different ways and magnitudes to a fruit being marketable or to its deterioration and, therefore, to its wastage. In my opinion, the authors need to explain what they were referring to, when they decided to include this measurement.

Author’s Response: The authors termed the trait “Wasted fruit (counted fruit per column)”. The term is explaining wastage as a partially developed fruit per fruit column. Wasted fruit is a loss to the overall productivity trait (while counting the number of marketable fruit per fruit column) and is under bimodal digenic inheritance as shown in Figure 1d.

The authors made changes according to the reviewer's suggestion for making clarity about the marketable fruit and wasted fruit. Please see below

Line 121-123: number of marketable fruit (counted per fruit column; refer to fully developed fruit), and number of wasted fruit (counted per fruit column; refers to partially developed fruit also known as carpelloid fruit)  

  1. In line 130 reference is made to Kanchana-udomkan et al., 2020, but in the list of references there is not this author and year.

Author’s Response: In Line 130 the reference (Kanchana-udomkan et al., 2020) has been added as [20].

Results:

  1. In lines 225 to 247 are listed the values obtained for the different characters measured in the different environments, but this information is also found in tables 4a and 4b. Why repeat the information? Only those results that were most genetically significant within the study should be emphasized.

Author’s Response: The authors emphasized reporting the key results from Line 225-247 of Table 4 a,b. Therefore, the authors would like to keep the results in a current form. Also, sections 3.3 and 3.4 were merged during editing in the journal system. I separated each section now.

  1. In table 4a, there are very high standard error values, specifically in the trait “Fruit weight (g)” (from ± 59.72 to ± 75.5) it indicates a greater dispersion of the data, and less precise confidence intervals.

Author’s Response: We bred the breeding lines of different fruit sizes (600-1200 g) as shown in Figure 1g according to the papaya industry's needs. Therefore, the acceptable range of standard errors was obtained because of the genotype selection for different fruit sizes across the RIL population. Also, within each RIL the standard error was very low, but we focused on trait stability across different environments fulfilling industry needs for picking, packing and transportation.  

  1. It is not possible to see the whole of Table 5, its dimensions exceed the margins of the manuscript.

Author’s Response: The authors fixed the margin of Table 5

Discussion:

  1. In lines 295-299 authors stated “Similarly, major environmental influences were previously demonstrated to affect the expression of fruit size (length and width), fruit weight and soluble solid contents (ºBrix) in apple (Malus pomila; Kumar et al., 2012), citrus (Citrus; Imai et al., 2018) and avocado (Persea americana; Cañas-Gutiérrez et al., 2022)”. I wonder if the high standard errors presented in table 4, for the case of Fruit weight can be explained only by the influence of the environment. Why, if there are several genetic studies carried out in different countries on papaya, on most of the characters studied here, do the authors refer to studies carried out on other crops, quite taxonomically distant from papaya, even if they are fruit trees?

Author’s Response: The current study focused on investigating the impact of two distinct but related agro-geographical climates in Tropical North Queensland; the Tablelands and Coastal regions on red papaya tree productivity and fruit quality traits. The data was collected from four trial sites, two trial sites each within each agro-geographical climate.

The authors emphasised the major environmental influences on the other tree crops where multi-environmental trials were focused whereas in papaya the trait stability analysis under distinct climatic effects is missing. Therefore, we compared papaya to other tree crops similar to the approach used in comparative genomics for gene expression and analysis.

  1. Line 318, reference Zulfi and Luo (2018) there is not in the reference list.

Author’s Response: The authors added the reference of Zulfi and Luo (2018) as [18; line 135].

  1. In general, the findings of this study are compared and discussed with the results obtained in Citrus and avocado, which are physiologically different plants from papaya and have longer crop cycles, and different agronomic management. If there were no studies of this type in papaya, the use of these references could be partly explained, but there are several previous studies in India and Brazil on the heritability and expression of many of the characters studied in this research.

Author’s Response: The studies focused on papaya in India and Brazil were confined to a single environment under which the plants were grown. Therefore, the authors cited the references from other tree crops like Citrus and Avocado (multi-environment studies) similar to the approach we adopt in comparative genomics. Multi-environmental trials are the best tool to assess the trait stability across selected elite genotypes for cultivar development.

References:

  1. The reference that appears with the number 9, Cañas-Gutierrez, this surname is misspelled in the list of references.

Author’s Response: The authors have fixed that.

  1. In general, there are references in the text that are not in the list of references at the end of the manuscript. And in the list of references there are 18 citations that are not in the text, these are the ones corresponding to the numbers: 3, 5, 6, 10, 15, 17, 18, 22, 24, 26, 35, 36, 37, 38, 39, 40, 41, 43.

Author’s Response: The authors have fixed that.

Reviewer 3 Report

Research questions are well defined and within the aims and the scope of the journal. Material is accordingly defined. Methods are suitable, properly described. The investigation is performed to good technical standards. It is no ethical problem involved. Conclusions are well stated and based on the results. Discussion is sound and relevant.
Suggestion:

Introduction part of the paper is too long.

Table 1 should be reorganised to be more clear (left column is not clear).

Table 4 should be reorganised to be more economic with the space and should be moved to the supplemental material.

Author Response

Research questions are well defined and within the aims and the scope of the journal. Material is accordingly defined.Methods are suitable and properly described. The investigation is performed to good technical standards. It is no ethicalproblem involved. Conclusions are well stated and based on the results. The discussion is sound and relevant.

Suggestions:

The introduction part of the paper is too long.

Author’s Response: The authors would like to keep the Introduction in the current form as it consists of only 531 words. The current size of the introduction is necessary to address the scope and goal of the current study to address the key issue of trait stabilization for the development of new red papaya varieties for the growing industry in Tropical North Queensland.

Table 1 should be reorganised to be more clear (left column is not clear).

Author’s Response: The authors are thankful to the reviewer for a deep look into our paper and for providing a minor suggestion for Table 1. Table 1 is about the Genotype (G) x Environment (E) interaction. The authors focused on trait stability by measuring variation (G X E) across the two distinct agro-geographical climates and within each climate each with two trail sites.

The authors are confident with the Table 1 description and layout for providing information on a genetic component influenced by each environment on measured traits across the F5 RIL population.

However, a minor change was done according to the reviewer's suggestion in the left column by replacing the “location” with “environment”

Table 4 should be reorganised to be more economic with the space and should be moved to the supplemental material.

Author’s Response: The authors would like to place Table 4 in the main manuscript due to its diligence and importance in explaining the genetic worth of each RIL using BLUP methods based on the REML approach.

Reviewer 4 Report

Ali and co-authors evaluated trait stability and inheritance pattern of red papaya genotypes in 4 sties and two ages with genotypes from a F5 RIL. The experiment was well designed with solid and rich data. The MS is well prepared and easy to follow with novel insights into the stability of key traits in response to various environmental conditions. The results will be beneficial for breeding new cultivars of red papaya. I only have two minor concerns:

It is rather surprising to see that the variance in fruit length is 0 for all conditions (Table 1)

Line 106, RIL seeds?

Author Response

Comments and Suggestions for Authors

Ali and co-authors evaluated trait stability and inheritance pattern of red papaya genotypes in 4 sties and two ages with genotypes from a F5 RIL. The experiment was well designed with solid and rich data. The MS is well prepared and easy to follow with novel insights into the stability of key traits in response to various environmental conditions. The results will be beneficial for breeding new cultivars of red papaya. I only have two minor concerns:

It is rather surprising to see that the variance in fruit length is 0 for all conditions (Table 1)

Author’s Response: The lines were bred for Tablelands and Coastal regions and had fruit sizes ranging between 600-1200 g (Figure 1g). Therefore, the breeding lines on the Coast and Tablelands had little variation in terms of fruit length (Table 1) and showed better stability across the two climates with minor variation in BLUP values for fruit length (Table 4 a,b).

Line 106, RIL seeds?

Author’s Response: The authors checked the grammar, and the correct version is “RIL seed”. The authors would like to keep the sentence as such (Line 106).

Round 2

Reviewer 2 Report

Comments and Suggestions for Authors

Manuscript ID: horticulturae-1883377

Title: The inheritance pattern of key desirable agronomic and fruit quality traits in elite red papaya genotypes.

 In my opinion this manuscript should be:  Accept after minor revision 

For the entire manuscript:

The authors addressed most of the aspects suggested, but some aspects still need to be improved:

-In Instructions for Authors, it specifies “References must be numbered in order of appearance in the text (including table captions and figure legends) and listed individually at the end of the manuscript”. In This case, there is no logical sequence of numbering them as they appear. This must be amended.

-Also the surname of the first author of reference 10 is misspelled, the correct surname is Cañas-Gutierrez.

-Regarding the discussion, although the authors certainly added several references to papaya, I believe some aspects of their study could have had a somewhat more extended discussion. There was also no explanation for the high standard errors obtained for the fruit weight trait.

Author Response

Second (2) Revision by Reviewer 2

The authors addressed most of the aspects suggested, but some aspects still need to be improved:

-In Instructions for Authors, it specifies “References must be numbered in order of appearance in the text (including table captions and figure legends) and listed individually at the end of the manuscript”. In This case, there is no logical sequence of numbering them as they appear. This must be amended.

Author’s Response: The authors have amended the references with numbers in a logical sequence as mentioned in the author's information. Please check the reference list (line 372-545)

-Also the surname of the first author of reference 10 is misspelt, the correct surname is CañasGutierrez.

Author’s Response: The authors corrected the name

Line 345: Corrected as “CañasGutierrez”

-Regarding the discussion, although the authors certainly added several references to papaya, I believe some aspects of their study could have had a somewhat more extended discussion. There was also no explanation for the high standard errors obtained for the fruit weight trait.

Author’s Response: The authors believe that the current discussion strongly supports the results presented in the current manuscript linked back to the previous literature and with new findings embedded in the current study. The cohort of the discussion covering key results strongly supports the novelty of the current study for the application in papaya breeding programs globally. The authors would like to thank the reviewer for a deep look into our manuscript. The authors corrected the standard errors presented in Table 4a due to typing mistakes. The correct version has been inserted in Table 4a (with track changes).